# Preconditioning Matters: Fast Global Convergence of Non-convex Matrix Factorization via Scaled Gradient Descent

**Xixi Jia**[1]  **Hailin Wang**[2]  **Jiangjun Peng**[2]  **Xiangchu Feng**[1]*  **Deyu Meng**[2,3]

[1]School of Mathematics and Statistics, Xidian University
[2]School of Mathematics and Statistics, Xi'an Jiaotong University
[3] Macao Institute of Systems Engineering, Macau University of Science and Technology
{hsijiaxidian, andrew.pengjj}@gmail.com; wanghailin97@163.com
xcfeng@mail.xidian.edu.cn; dymeng@xjtu.edu.cn

## Abstract

Low-rank matrix factorization (LRMF) is a canonical problem in non-convex optimization, the objective function to be minimized is non-convex and even non-smooth, which makes the global convergence guarantee of gradient-based algorithm quite challenging. Recent work made a breakthrough on proving that standard gradient descent converges to the $\varepsilon$-global minima after $O(\frac{d\kappa^2}{\tau^2}\ln\frac{d\sigma_d}{\tau} + \frac{d\kappa^2}{\tau^2}\ln\frac{\sigma_d}{\varepsilon})$ iterations from small initialization with a very small learning rate (both are related to the small constant $\tau$). While the dependence of the convergence on the *condition number* $\kappa$ and *small learning rate* makes it not practical especially for ill-conditioned LRMF problem.

In this paper, we show that precondition helps in accelerating the convergence and prove that the scaled gradient descent (ScaledGD) and its variant, alternating scaled gradient descent (AltScaledGD) converge to an $\varepsilon$-global minima after $O(\ln\frac{d}{\delta}+\ln\frac{d}{\varepsilon})$ iterations from general random initialization. Meanwhile, for small initialization as in gradient descent, both ScaledGD and AltScaledGD converge to $\varepsilon$-global minima after only $O(\ln\frac{d}{\varepsilon})$ iterations. Furthermore, we prove that as a proximity to the alternating minimization, AltScaledGD converges faster than ScaledGD, its global convergence does not rely on small learning rate and small initialization, which certificates the advantages of AltScaledGD in LRMF.

## 1 Introduction

Low-rank matrix factorization aims to approximate a given rank $d$ matrix $\boldsymbol{M} \in \mathbb{R}^{m \times n}$ by the product of two factor matrices $\boldsymbol{U} \in \mathbb{R}^{m \times d}, \boldsymbol{V} \in \mathbb{R}^{n \times d}$, which plays a fundamental and essential role in low-rank matrix recovery such as matrix completion Jain et al. [2013], Ge et al. [2016], Sun and Luo [2016], matrix sensing Chi et al. [2019], Zhao et al. [2015], Charisopoulos et al. [2021], robust principal component analysis Candès et al. [2011], Cai et al. [2021], and the theoretical analysis of deep neural network Du et al. [2018]. Meanwhile, low-rank matrix factorization is also viewed as a canonical problem in non-convex optimization as the objective function to be minimized is non-convex and even non-smooth. Mathematically, we are to solve

$$\min_{\boldsymbol{U}\in\mathbb{R}^{m\times d},\boldsymbol{V}\in\mathbb{R}^{n\times d}} f(\boldsymbol{U},\boldsymbol{V}) := \frac{1}{2}\|\boldsymbol{U}\boldsymbol{V}^\top - \boldsymbol{M}\|_F^2, \tag{1}$$

---

*Corresponding author.

37th Conference on Neural Information Processing Systems (NeurIPS 2023).

where $d \ll \min(m, n)$. Though problem (1) is not difficult to solve, the study of this problem has great significance to the gradient-based algorithm for low-rank matrix recovery Hou et al. [2020], Li et al. [2019b], Chen et al. [2019], Tong et al. [2021], as it is exactly the population loss of low-rank matrix recovery models. Meanwhile, from the perspective of non-convex optimization, problem (1) is an ideal test bed for the theoretical analysis of the asymptotic convergence of gradient-based algorithm for non-convex optimization.

The theoretical guarantee for the global convergence of gradient-based algorithm for problem (1) is challenging, which is due to the following reasons: 1) the problem is non-convex with respect to the variables $U$ and $V$, and there are infinitely many local minima and saddle points. Specifically, if $U^*$ and $V^*$ is an optimal solution of problem (1), then $U^*Q$ and $V^*Q^{-\top}$ is also an optimal solution for any invertible matrix $Q$; 2) the problem is non-smooth with respect to the variables $U$ and $V$ and is not coercive due to $f(\alpha U \frac{1}{\alpha} V^\top) = f(UV^\top)$ where the scalar $\alpha$ can be arbitrarily large or small. In theory, gradient-based algorithm is only able to find critical points, while practically, gradient descent algorithm has been verified to converge to the global minima of problem (1) efficiently.

To close the gap between theory and practice, Li et al. [2019a], Ge et al. [2017], Chi et al. [2019] proved that even-though the loss in Eq. (1) is non-convex its loss landscape has some nice property: all local minima are global optima and all the saddle points are strict saddles. Therefore gradient descent algorithms can be guarantee to converge to the global minima. To help escape the strict saddles, Jin et al. [2017] proposed perturbed gradient descent by adding isotropic noise to the gradient at each iteration, they prove that with high probability, perturbed gradient descent converges to the global minima from random initialization at a linear rate. While as analyzed in Ye and Du [2021] and verified by experiments that the gradient perturbation is not really necessary for problem (1) .

In contrast to the perturbed gradient descent, Du et al. [2018] studied the naive gradient descent for problem (1), they exploited the balancedness of the two factors $\|U\|_F^2 - \|V\|_F^2$ maintained by gradient flow and proved polynomial convergence rate of gradient descent for problem (1) when $d = 1$. Furthermore, Ye and Du [2021] improved the results of Du et al. [2018] to rank $d$ case, they proved that gradient descent converges to the $\varepsilon$-global minima of problem (1) after $O(\frac{d\kappa^2}{\tau^2}\ln\frac{d\sigma_d}{\tau} + \frac{d\kappa^2}{\tau^2}\ln\frac{\sigma_d}{\varepsilon})$ iterations. Ye and Du [2021] divided the convergence process into two stages: warm-up phase which takes $O(\frac{d\kappa^2}{\tau^2}\ln\frac{d\sigma_d}{\tau})$ iterations and local convergence phase which takes $O(\frac{d\kappa^2}{\tau^2}\ln\frac{\sigma_d}{\varepsilon})$ iterations. The warm-up phase in Ye and Du [2021] is actually the saddle avoid phase after which the gradient descent escapes all the saddle regions as shown in Fig. 1.

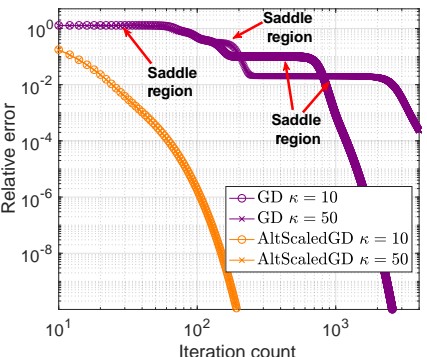

Figure 1: Illustration of the global convergence of GD and AltScaledGD from random initialization.

It can be seen from Fig. 1 and proved by Ye and Du [2021] that both the saddle avoid phase and local convergence phase of gradient descent highly rely on the condition number $\kappa$ of the matrix $M$. If the condition number $\kappa$ is large, then gradient descent takes long time to escape the saddle regions and also converges slowly. It is therefore very important to know *can we improve the gradient-based algorithm such that the global convergence is independent of the condition number?* Besides, the global convergence of both Du et al. [2018] and Ye and Du [2021] require very small learning rate (related to the small constant $\tau$), which seriously limits the application of gradient descent algorithm for ill-conditioned LRMF problem.

Recently, the scaled gradient descent algorithm (ScaledGD) Apuroop [2012], Mishra and Sepulchre [2016], Tanner and Wei [2016], has been proved by Tong et al. [2021], Tong [2022] to converge very fast from specialized initialization (spectral initialization) to the global minima of problem (1) and the convergence rate is independent of the condition number (Theorem 5 in Tong et al. [2021]). Yet the convergence result provided by Tong et al. [2021] is only local, whether the scaled gradient algorithm can escape saddle regions efficiently for the non-convex problem (1) is still not clear.

In this paper, we are the first to prove that ScaledGD as well as AltScaledGD converge to the global minima of problem (1) from general random Gaussian initialization, and the convergence rate is *independent of the condition number* of the matrix $M$. Moreover, we show that the global convergence

results of ScaledGD and AltScaledGD *do not rely on small initialization*, the global convergence of AltScaledGD *does not even require a small learning rate*, which significantly improves the result of Ye and Du [2021]. To sum up, the contributions of this paper are three-fold:

1. We provide a very simple proof framework for the convergence of ScaledGD and AltScaledGD from general random initialization. Specifically, we divide the optimization process into three phases: **initial phase, saddle avoid phase and linear convergence phase**. We prove that the loss decreases at rate $(1 - \eta)^{2k}$ in the initial phase, and further decreases linearly as $(1 - \chi_k)^k$ in the saddle avoid phase and the linear convergence phase, where $\eta$ and $\chi_k$ are independent of the condition number $\kappa$ and $\chi_k$ is monotonically increasing from $\frac{\eta^2}{(2-\eta)^2}$ to $\eta$;

2. We prove that if the scale of the random initialization is smaller than a given constant (small initialization), then the loss decrease linearly as $(1 - \eta)^k$ from such small initialization to the global minima for both ScaledGD and AltScaledGD.

3. We show that AltScaledGD is a significant improvement of the ScaledGD in that it converges fast with large learning rate up to 1. While in contrast the learning rate of ScaledGD should be smaller than a constant $c_\eta$ that is much smaller than 1.

The organization of this paper is as follow. In Section 2, we introduce the related works on ScaledGD and AltScaledGD. In Section 3, we present our main results, then we give more detailed theoretical analysis on the proof sketch in Section 4. Finally, we conclude this work in Section 5.

## 2 Related work

In this section, we introduce the ScaledGD and the AltScaledGD as specified in Apuroop [2012], Mishra and Sepulchre [2016], Tanner and Wei [2016], Tong et al. [2021]. We show that our work is a significant improvement to these existing works on the global convergence analysis.

### 2.1 Scaled gradient descent

Different to the gradient descent algorithm which takes the negative gradient direction as the descent direction, scaled gradient descent is designed to accelerate the convergence process by scaled the gradient with a preconditioning matrix. Specifically, the ScaledGD updates the variables $\boldsymbol{U}_k, \boldsymbol{V}_k$ as

$$
\begin{cases}
\boldsymbol{U}_{k+1} = \boldsymbol{U}_k - \eta \nabla_{\boldsymbol{U}_k} f(\boldsymbol{U}_k, \boldsymbol{V}_k) \left(\boldsymbol{V}_k^\top \boldsymbol{V}_k\right)^{-1} \\
\boldsymbol{V}_{k+1} = \boldsymbol{V}_k - \eta \nabla_{\boldsymbol{V}_k} f(\boldsymbol{U}_k, \boldsymbol{V}_k) \left(\boldsymbol{U}_k^\top \boldsymbol{U}_k\right)^{-1}
\end{cases}
\tag{2}
$$

where $\eta$ is the learning rate, $\boldsymbol{U}_k^\top \boldsymbol{U}_k$ and $\boldsymbol{V}_k^\top \boldsymbol{V}_k$ are matrices of $d \times d$, and $d \ll \min(m, n)$ therefore the computation of ScaledGD is comparable to that of gradient descent. The inverse matrices $\left(\boldsymbol{V}_k^\top \boldsymbol{V}_k\right)^{-1}$ and $\left(\boldsymbol{U}_k^\top \boldsymbol{U}_k\right)^{-1}$ is the preconditioning for the gradient descent. If we denote $\boldsymbol{X} = (\boldsymbol{U}, \boldsymbol{V})$, then Eq. (2) corresponds to

$$
\boldsymbol{X}_{k+1} = \boldsymbol{X}_k - \eta \nabla f_{\boldsymbol{X}_k}(\boldsymbol{X}_k) \boldsymbol{H}_k
\tag{3}
$$

where $\boldsymbol{H}_k = \begin{bmatrix} \left(\boldsymbol{V}_k^\top \boldsymbol{V}_k\right)^{-1} & 0 \\ 0 & \left(\boldsymbol{U}_k^\top \boldsymbol{U}_k\right)^{-1} \end{bmatrix}$.

Apuroop [2012], Mishra and Sepulchre [2016] proved that ScaledGD is derived by imposing a new metric on the tangent space of the Riemannian manifold. They verified empirically that ScaledGD converges much faster than gradient descent while there is no rigious convergence rate analysis. Recently, Tong et al. [2021] is the first to prove the linear convergence property of Eq. (2) for problem (1), while their proof relies on specialized initialization that $\text{dist}(\boldsymbol{X}_0, \boldsymbol{X}_*) \leq 0.1\sigma_d(\boldsymbol{M})$, where $\boldsymbol{X}^* = (\boldsymbol{U}^*, \boldsymbol{V}^*)$ and $\boldsymbol{U}^*\boldsymbol{V}^{*\top} = \boldsymbol{M}$ (Theorem 5 Tong et al. [2021]). The local convergence guarantee is far from satisfactory to understand the convergence of ScaledGD for the non-convex optimization problem (1).

## 2.2 Alternating scaled gradient descent

The Gaussian-Seidel version of ScaledGD is the following alternating scaled gradient descent which writes

$$\begin{cases} \boldsymbol{U}_{k+1} = \boldsymbol{U}_k - \eta \nabla_{\boldsymbol{U}_k} f(\boldsymbol{U}_k, \boldsymbol{V}_k)(\boldsymbol{V}_k^\top \boldsymbol{V}_k)^{-1} \\ \boldsymbol{V}_{k+1} = \boldsymbol{V}_k - \eta \nabla_{\boldsymbol{V}_k} f(\boldsymbol{U}_{k+1}, \boldsymbol{V}_k)(\boldsymbol{U}_{k+1}^\top \boldsymbol{U}_{k+1})^{-1} \end{cases} \tag{4}$$

Eq. (4) was studied as scaled alternating steepest descent algorithm in Tanner and Wei [2016], and it is also closely related to the alternating minimization algorithm for the minimization problem (1) when $\eta = 1$, and other matrix recovery problem as Wen et al. [2012], Jain et al. [2013], Chandrasekher et al. [2022]. Different to alternating minimization, the AltScaledGD presented in Eq. (4) can be broadly used in lots of low-rank matrix recovery problem where alternating minimization is computationally prohibitive, such as matrix completion Zilber and Nadler [2022], Sun and Luo [2016], matrix sensing Ma et al. [2021]. Existing works for Eq. (4) only proved convergence to critical point as Wen et al. [2012], Tanner and Wei [2016], yet global convergence analysis of AltScaledGD is still vague.

In this paper, we provide rigorous proofs for the global convergence of ScaledGD Eq. (2) and AltScaledGD Eq. (4), and we show that both ScaledGD and AltScaledGD converge linearly for random Gaussian initialization after saddle avoid phase. Meanwhile, we show that AltScaledGD is robust to the learning rate $\eta$ which can be set as large as 1, while large $\eta$ can seriously deteriorates the convergence property of ScaledGD as illustrated by Fig. 6, which sheds light on the superiority of AltScaledGD Eq. (4) over ScaledGD Eq. (2) on problem (1) as well as more low-rank matrix recovery problem.

## 3 Main results

In this section, we present our main theorems on the convergence of ScaledGD and AltScaledGD for two different random initialization: general random initialization and small initialization. These two initializations are both random Gaussian initialization with zero mean but different variances. Small initialization is widely used in the convergence analysis of low rank matrix factorization problem Stöger and Soltanolkotabi [2021], Ye and Du [2021], Ma and Fattahi [2022], while small initialization is skin to spectral initialization which does not help us fully understand the global convergence of the non-convex problem. In this paper, we provide both the global convergence analysis of general random initialization and small initialization.

### 3.1 Global convergence of ScaledGD

If the matrix $\boldsymbol{M}$ is rank one, i.e., $d$ in Eq. (1) is 1, then Eq. (2) is exactly gradient descent with adaptive step-size. We show that such specialized gradient descent for $d \geq 1$ converges linearly to the global minima after an initial decreasing phase and the convergence rate is independent of the singular value of $\boldsymbol{M}$.

**Theorem 1** (General random initialization). *Let $\boldsymbol{U}_0 \in \mathbb{R}^{m \times d}$ and $\boldsymbol{V}_0 \in \mathbb{R}^{n \times d}$ be random Gaussian that follow $\mathcal{N}(0, \sigma)$ for $\sigma > c_{\mathrm{init}}$ ($c_{\mathrm{init}}$ is a positive constant), and $\boldsymbol{U}_k$, $\boldsymbol{V}_k$ are updated by Eq. (2). If $\eta \leq c_\eta < 1$ for small constant $c_\eta$, we have that the objective function of problem (1) decreases linearly after $T_1 = O\left(\ln \frac{d}{\delta}\right)$ iterations, namely*

$$\|\boldsymbol{U}_{k+T_1} \boldsymbol{V}_{k+T_1}^\top - \boldsymbol{M}\|_F \leq \alpha_1 \left(1 - \chi_{k+T_1}\right)^k \|\boldsymbol{M}\|_F, \forall k \geq 0 \tag{5}$$

*where $\chi_{k+T_1}$ is monotonically increasing from $\frac{\eta^2}{(2-\eta)^2}$ to $\eta$, $\delta$ is a sufficiently small constant, $\alpha_1$ is a constant.*

The Theorem 1 indicates that the global convergence of ScaledGD can be divided into three phases: the **initial phase** that lasts $T_1$ iterations, the **saddle avoid phase** in which $\chi_{k+T_1}$ increases from $\frac{\eta/2}{1-\eta/2}$ to $\eta$ and the final **linear convergence phase** with convergence rate $1 - \eta$. While if the scale of the initialization $\boldsymbol{U}_0$ and $\boldsymbol{V}_0$ are very small (with small $\sigma$), then the following theorem shows that the ScaledGD converges linearly without entering the saddle regions.

**Theorem 2** (Small initialization). *Let $\boldsymbol{U}_0 \in \mathbb{R}^{m \times d}$ and $\boldsymbol{V}_0 \in \mathbb{R}^{n \times d}$ be random Gaussian that follow $\mathcal{N}(0, \sigma)$, with $\sigma \leq c_{\mathrm{init}}$ and $\boldsymbol{U}_k$, $\boldsymbol{V}_k$ are updated by Eq. (2). If $\eta \leq c_\eta < 1$ for small constant $c_\eta$, we have that the objective function of problem (1) decreases linearly, namely*

$$\|\boldsymbol{U}_k \boldsymbol{V}_k^\top - \boldsymbol{M}\|_F \leq \alpha_2 \left(1 - \eta\right)^k \|\boldsymbol{M}\|_F \tag{6}$$

*where $c_{\text{init}}$ is a small constant and $\alpha_2$ is a constant.*

## 3.2 Global convergence of AltScaledGD

We now present the main convergence results of AltScaledGD.

**Theorem 3** (General random initialization). *Let $\boldsymbol{U}_0 \in \mathbb{R}^{m \times d}$ and $\boldsymbol{V}_0 \in \mathbb{R}^{n \times d}$ be random Gaussian that follow $\mathcal{N}(0, \sigma)$ for $\sigma > c_{\text{init}}$, $\boldsymbol{U}_k$, $\boldsymbol{V}_k$ are updated by Eq. (4), we have that the objective function of problem (1) decreases linearly after $T_1 = O(\ln \frac{d}{\delta})$ iterations, namely*

$$\|\boldsymbol{U}_{k+T_1} \boldsymbol{V}_{k+T_1}^\top - \boldsymbol{M}\|_F \le \alpha_1 \left(1 - \chi_{k+T_1}\right)^k \|\boldsymbol{M}\|_F \tag{7}$$

*where $\chi_{k+T_1}$ is monotonically increasing from $\frac{\eta^2}{(2-\eta)^2}$ to $\eta$, $0 < \eta \le 1$ and $\alpha_1$ is a constant.*

**Theorem 4** (Small initialization). *Let $\boldsymbol{U}_0 \in \mathbb{R}^{m \times d}$ and $\boldsymbol{V}_0 \in \mathbb{R}^{n \times d}$ be random Gaussian that follow $\mathcal{N}(0, \rho)$, with $\sigma \le c_{\text{init}}$, $\boldsymbol{U}_k$, $\boldsymbol{V}_k$ are updated by Eq. (4) then we have that the objective function of problem (1) decreases linearly, namely*

$$\|\boldsymbol{U}_k \boldsymbol{V}_k^\top - \boldsymbol{M}\|_F \le \alpha_2 \left(1 - \eta\right)^k \|\boldsymbol{M}\|_F \tag{8}$$

*where $0 < \eta \le 1$ is the step size, $\alpha_2$ is a constant and $c_{\text{init}}$ is a small constant.*

The convergence results of ScaledGD and AltScaledGD are almost the same with differences in that for ScaledGD the learning $\eta$ should be smaller than a constant $c_\eta$ which is much less than 1. The small constant $c_\eta$ greatly restricts the convergence rate of ScaledGD. While for AltScaledGD the learning rate $\eta$ can be as large as 1, which indicates the superiority of AltScaledGD over ScaledGD in convergence as $\eta$ is crucial to the convergence rate. From the above theorems, it can be easily deduced that both ScaledGD and AltScaledGD converge to an $\varepsilon$-global minima after $O(\ln \frac{d}{\delta} + \ln \frac{d}{\varepsilon})$ iterations from general random initialization. While for small initialization, these two algorithms only need $O(\ln \frac{d}{\varepsilon})$ iterations to converge to an $\varepsilon$-global minima. More detailed analysis and proofs are provided in Section 4 the proof sketch part and the supplementary materials.

## 3.3 Why does preconditioning help?

In previous works, preconditioning has been used to improve the condition of the optimization problem Saad [2003], Zhang et al. [2021]. In this paper, we analyze how does the preconditioning help improving the convergence by analyzing the effect of condition number $\kappa$ on the learning rate $\eta$ as the convergence rate highly depends on the learning rate. For simplicity, we take one step of the AltScaledGD as example to analyze the effect of the preconditioning. Since $f(\boldsymbol{U}, \boldsymbol{V})$ is quadratic on $\boldsymbol{U}$, then it can be verified

$$f(\boldsymbol{U}_{k+1}, \boldsymbol{V}_k) \le f(\boldsymbol{U}_k, \boldsymbol{V}_k) + \langle \nabla f_{\boldsymbol{U}}, \Delta \rangle + \frac{1}{2} \|\Delta\|_{\mathcal{A}_k}^2 \tag{9}$$

where $\Delta = \boldsymbol{U}_{k+1} - \boldsymbol{U}_k$ and $\|\cdot\|_{\mathcal{A}_k}$ is a local norm defined by $\|\Delta\|_{\mathcal{A}_k} = \langle \Delta \boldsymbol{V}_k^\top \boldsymbol{V}_k, \Delta \rangle$.

For gradient descent, we take $\Delta = -\eta \nabla f_{\boldsymbol{U}}(\boldsymbol{U}_k, \boldsymbol{V}_k)$, then Eq. (9) becomes

$$\begin{aligned}
f(\boldsymbol{U}_{k+1}, \boldsymbol{V}_k) &\le f(\boldsymbol{U}_k, \boldsymbol{V}_k) - \eta \|\nabla f_{\boldsymbol{U}}(\boldsymbol{U}_k, \boldsymbol{V}_k)\|_F^2 + \frac{\eta^2 \sigma_1^2(\boldsymbol{V}_k)}{2} \|\nabla f_{\boldsymbol{U}}(\boldsymbol{U}_k, \boldsymbol{V}_k)\|_F^2 \\
&\le f(\boldsymbol{U}_k, \boldsymbol{V}_k) - 2\eta \sigma_d^2(\boldsymbol{V}_k) f(\boldsymbol{U}_k, \boldsymbol{V}_k) + \eta^2 \sigma_1^4(\boldsymbol{V}_k) f(\boldsymbol{U}_k, \boldsymbol{V}_k) \\
&\le \left(1 - (2\eta \sigma_d^2(\boldsymbol{V}_k) - \eta^2 \sigma_1^4(\boldsymbol{V}_k))\right) f(\boldsymbol{U}_k, \boldsymbol{V}_k)
\end{aligned} \tag{10}$$

Similarly, it holds

$$f(\boldsymbol{U}_{k+1}, \boldsymbol{V}_{k+1}) \le \left(1 - (2\eta \sigma_d^2(\boldsymbol{U}_{k+1}) - \eta^2 \sigma_1^4(\boldsymbol{U}_{k+1}))\right) f(\boldsymbol{U}_{k+1}, \boldsymbol{V}_k) \tag{11}$$

Therefore, we have

$$f(\boldsymbol{U}_{k+1}, \boldsymbol{V}_{k+1}) \le \left(1 - (2\eta \sigma_d^2(\boldsymbol{U}_{k+1}) - \eta^2 \sigma_1^4(\boldsymbol{U}_{k+1}))\right) \left(1 - (2\eta \sigma_d^2(\boldsymbol{V}_k) - \eta^2 \sigma_1^4(\boldsymbol{V}_k))\right) f(\boldsymbol{U}_k, \boldsymbol{V}_k) \tag{12}$$

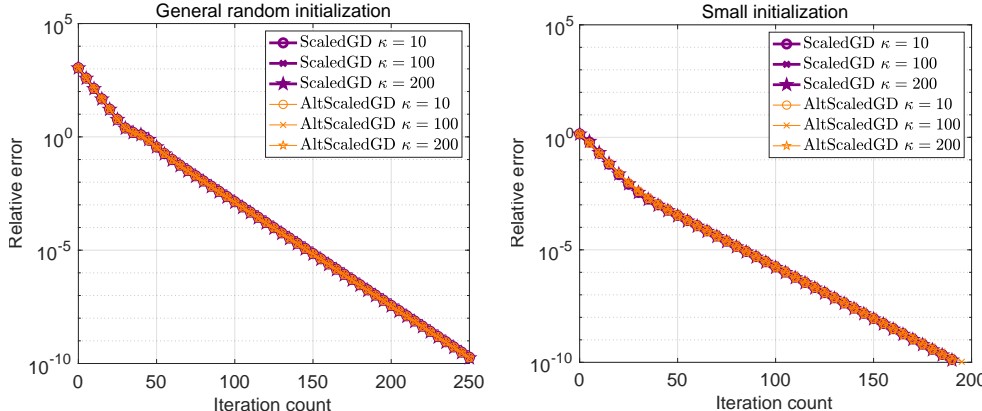

Figure 2: Illustration of convergence of ScaledGD and AltScaledGD under different condition $\kappa$ and different initialization.

To guarantee the linear convergence of gradient descent, it is required that $2\eta\sigma_r^2(\boldsymbol{V}_k) \geq \eta^2\sigma_1^4(\boldsymbol{V}_k)$ and $2\eta\sigma_r^2(\boldsymbol{U}_{k+1}) \geq \eta^2\sigma_1^4(\boldsymbol{U}_{k+1})$ which implies $\eta \leq \min\{\frac{2}{\sigma_1^2(\boldsymbol{V}_k)}\kappa(\boldsymbol{V}_k)^2, \frac{2}{\sigma_1^2(\boldsymbol{U}_{k+1})}\kappa(\boldsymbol{U}_{k+1})^2\}$ [2]. In contrast, if we take $\Delta = -\eta\nabla f_{\boldsymbol{U}}(\boldsymbol{U}_k, \boldsymbol{V}_k)(\boldsymbol{V}_k^\top\boldsymbol{V}_k)^{-1}$, then

$$f(\boldsymbol{U}_{k+1}, \boldsymbol{V}_k) \leq f(\boldsymbol{U}_k, \boldsymbol{V}_k) - (\eta - \frac{\eta^2}{2})\left\langle(\boldsymbol{U}_k\boldsymbol{V}_k^\top - \boldsymbol{M})\mathcal{V}_k\mathcal{V}_k^\top, (\boldsymbol{U}_k\boldsymbol{V}_k^\top - \boldsymbol{M})\right\rangle$$

$$\leq f(\boldsymbol{U}_k, \boldsymbol{V}_k) - (\eta - \frac{\eta^2}{2})\sigma_r\left(\mathcal{V}_E^\top\mathcal{V}_k\right)f(\boldsymbol{U}_k, \boldsymbol{V}_k) \qquad (13)$$

$$= \left(1 - (\eta - \frac{\eta^2}{2})\sigma_r\left(\mathcal{V}_E^\top\mathcal{V}_k\right)\right)f(\boldsymbol{U}_k, \boldsymbol{V}_k)$$

and similarly

$$f(\boldsymbol{U}_{k+1}, \boldsymbol{V}_{k+1}) \leq f(\boldsymbol{U}_{k+1}, \boldsymbol{V}_k) - (\eta - \frac{\eta^2}{2})\left\langle(\boldsymbol{V}_k\boldsymbol{U}_{k+1}^\top - \boldsymbol{M}^\top)\mathcal{U}_{k+1}\mathcal{U}_{k+1}^\top, (\boldsymbol{U}_{k+1}\boldsymbol{V}_k^\top - \boldsymbol{M})\right\rangle$$

$$\leq f(\boldsymbol{U}_{k+1}, \boldsymbol{V}_k) - (\eta - \frac{\eta^2}{2})\sigma_r\left(\mathcal{U}_E^\top\mathcal{U}_{k+1}\right)f(\boldsymbol{U}_{k+1}, \boldsymbol{V}_k) \qquad .$$

$$= \left(1 - (\eta - \frac{\eta^2}{2})\sigma_r\left(\mathcal{U}_E^\top\mathcal{U}_{k+1}\right)\right)f(\boldsymbol{U}_{k+1}, \boldsymbol{V}_k)$$

$$(14)$$

Thus

$$f(\boldsymbol{U}_{k+1}, \boldsymbol{V}_{k+1}) \leq \left(1 - (\eta - \frac{\eta^2}{2})\sigma_r\left(\mathcal{V}_E^\top\mathcal{V}_k\right)\right)\left(1 - (\eta - \frac{\eta^2}{2})\sigma_r\left(\mathcal{U}_E^\top\mathcal{U}_{k+1}\right)\right)f(\boldsymbol{U}_k, \boldsymbol{V}_k). \quad (15)$$

To guarantee the linear convergence, we only need $0 < \eta < 2$. $\sigma_r\left(\mathcal{U}_E^\top\mathcal{U}_{k+1}\right)$ as well as $\sigma_r\left(\mathcal{V}_E^\top\mathcal{V}_k\right)$ is strictly larger than 0 ($\mathcal{V}_E$ is the orthogonal row subspace of $\boldsymbol{U}_k\boldsymbol{V}_k^\top - \boldsymbol{M}$ and $\mathcal{V}_k$ is the orthogonal subspace of $\boldsymbol{V}_k$), which indicates that the linear convergence rate is independent of the condition number of matrix $\boldsymbol{M}$.

We show in Fig. 2 that the convergence of ScaledGD and AltScaledGD are independent of the condition number $\kappa$ of the matrix $\boldsymbol{M}$ with general random initialization and small initialization. In Fig. 2, we set the rank of the matrix $\boldsymbol{M}$ as 5, with condition number $\kappa$ ranging from $10, 100, 200$. It can be seen from the left subfigure of Fig. 2 that for general random initialization, the error curves of ScaledGD with different $\kappa$ are the exactly the same, and the error curves of ScaledGD also coincide with that of the AltScaledGD. These results are also true for small initialization as shown in the right subfigure of Fig. 2. These observations certificate that preconditioning in Eq. (2) and Eq. (4) indeed help accelerating the convergence such that the convergence rate is independent of the condition number of the matrix $\boldsymbol{M}$.

---

[2]Since $\boldsymbol{U}_k\boldsymbol{V}_k^\top \to \boldsymbol{M}, k \to \infty$ and the same analysis is applied on Eq. (10) with respect to $\boldsymbol{V}$, we know that $\eta \leq c\kappa$.

# 4 Theoretical analysis – proof sketch

In this section, we provide the proof sketch of our results in Section 3. For simplicity, we present the theoretical analysis for rank one matrix factorization where $U \in \mathbb{R}^{m \times 1}$ and $V \in \mathbb{R}^{n \times 1}$. More detailed proofs for the main theorems are provided in the supplementary material.

## 4.1 Convergence of ScaledGD

It can be deduced that the objective function of problem (1) is upper bounded by four terms as

$$\|U_{k+1}V_{k+1}^{\top} - M\|_F \leq \underbrace{(1-\eta)^2\|U_kV_k^{\top} - M\|_F}_{①} + (1-\eta)\eta\underbrace{\|M\|_F\|\mathcal{V}_{*\perp}^{\top}\mathcal{V}_k\|_2}_{②}$$

$$+ \eta(1-\eta)\underbrace{\|M\|_F\|\mathcal{U}_{*\perp}^{\top}\mathcal{U}_k\|_2}_{③} + \eta^2\underbrace{\|M\|_F\left|1 - \frac{\|M\|_F\cos\boldsymbol{\theta}_k^u\cos\boldsymbol{\theta}_k^v}{\|U_kV_k^{\top}\|_F}\right|}_{④} \quad (16)$$

where $\mathcal{U}_k$ and $\mathcal{V}_k$ correspond to the orthogonal basis of the column space of $U_k$ and $V_k$, $\mathcal{U}_{*\perp}$ and $\mathcal{V}_{*\perp}$ are the orthogonal complements of the left and right singular vector matrices of $M$ (i.e. $U_*$, $V_*$), and $\cos\boldsymbol{\theta}_k^u$ is cosine value of the angle between the vectors $U_k$ and $U_*$, $\cos\boldsymbol{\theta}_k^v$ is cosine value of the angle between the vectors $V_k$ and $V_*$. The upper-bound depicts the differences between $U_{k+1}V_{k+1}^{\top}$ and $M$ in two aspects

  ✧ The angle between the subspace of $U_kV_k^{\top}$ and $M$: $\|\mathcal{U}_{*\perp}^{\top}\mathcal{U}_k\|_2$, $\|\mathcal{V}_{*\perp}^{\top}\mathcal{V}_k\|_2$;

  ✧ The difference of the length (norm) between $U_kV_k^{\top}$ and $M$: $\|M\|_F - \|U_kV_k^{\top}\|_F$.

The term ② and ③ are related to the angle between the subspace of $M$ and $U_kV_k^{\top}$, the term ④ is related to the difference between the norm of $M$ and $U_kV_k^{\top}$, as given by the following lemma.

**Lemma 1.** *If* $\langle M, U_kV_k^{\top}\rangle \geq \|U_kV_k^{\top}\|_F^2$, *then there is constant* $C_u \geq 0$ *such that*

$$\left|1 - \frac{\|M\|_F\cos\boldsymbol{\theta}_k^u\cos\boldsymbol{\theta}_k^v}{\|U_kV_k^{\top}\|_F}\right| \leq C_u\left(\|M\|_F - \|U_kV_k^{\top}\|_F\right) \quad (17)$$

According to Eq. (16), we know that the decrease of the objective function in problem (1) is decided by the decrease of the distance between the subspace (② and ③) and the difference between the norm of $M$ and $U_kV_k^{\top}$ (④). The following lemma further reveals that the distance between the subspace of $U_kV_k^{\top}$ and $M$ decreases.

**Lemma 2.** (Convergence of the distance between subspaces) *For the ScaledGD (2), if* $\|M\|_F\cos\boldsymbol{\theta}_k^u\cos\boldsymbol{\theta}_k^v \geq \|U_kV_k^{\top}\|_F$, *then the following holds*

$$\|\mathcal{U}_{*\perp}^{\top}\mathcal{U}_{k+1}\|_2 \leq (1-\eta)\|\mathcal{U}_{*\perp}^{\top}\mathcal{U}_k\|_2, \quad \|\mathcal{V}_{*\perp}^{\top}\mathcal{V}_{k+1}\|_2 \leq (1-\eta)\|\mathcal{V}_{*\perp}^{\top}\mathcal{V}_k\|_2 \quad (18)$$

The Lemma 2 indicates that the term ② and ③ in Eq. (16) decrease linearly if the norm of $\|U_kV_k^{\top}\|_F$ is smaller than norm of the projection of $M$ onto the column and row spaces of $U_kV_k^{\top}$. At the mean time, the condition $\|M\|_F\cos\boldsymbol{\theta}_k^u\cos\boldsymbol{\theta}_k^v \geq \|U_kV_k^{\top}\|_F$ also guarantees the linear convergence of the differences between the norm of $U_kV_k^{\top}$ and $M$.

**Theorem 5.** (Convergence of the matrix norm) *For the ScaledGD (2), if* $\|M\|_F\cos\boldsymbol{\theta}_k^u\cos\boldsymbol{\theta}_k^v \geq \|U_kV_k^{\top}\|_F$ *for all* $k \geq 0$, *then we have*

$$\|M\|_F - \|U_{k+1}V_{k+1}^{\top}\|_F \leq (1-\eta)^{2k}kC_{\alpha} \quad (19)$$

*where* $C_{\alpha}$ *is a constant and* $\eta$ *is the step length* $0 \leq \eta < 1$.

Both Lemma 2 and Theorem 5 are built on the condition that $\|M\|_F\cos\boldsymbol{\theta}_k^u\cos\boldsymbol{\theta}_k^v \geq \|U_kV_k^{\top}\|_F$ for all $k \geq 0$, while this is not a trivial condition for ScaledGD. The following lemma guarantees that the condition can be satisfied if the step length $\eta$ is smaller than a constant.

**Lemma 3.** *Let* $\eta \leq c_{\eta} < 1$ *with* $c_{\eta}$ *a small constant, if* $\|M\|_F\cos\boldsymbol{\theta}_0^u\cos\boldsymbol{\theta}_0^v \geq \|U_0V_0^{\top}\|_F$ *then the following is true*

$$\|M\|_F\cos\boldsymbol{\theta}_k^u\cos\boldsymbol{\theta}_k^v \geq \|U_kV_k^{\top}\|_F, \forall k > 0. \quad (20)$$

The above results guarantee the local linear convergence of the term ②, ③ and ④ on the condition that

$$\|\boldsymbol{M}\|_F \cos\boldsymbol{\theta}_0^u \cos\boldsymbol{\theta}_0^v \geq \|\boldsymbol{U}_0\boldsymbol{V}_0^\top\|_F \tag{21}$$

which is critical for our analysis on random Gaussian initialization and small initialization.

### 4.1.1 Small initialization

In practice, the condition $\|\boldsymbol{M}\|_F \cos\boldsymbol{\theta}_0^u \cos\boldsymbol{\theta}_0^v \geq \|\boldsymbol{U}_0\boldsymbol{V}_0^\top\|_F$ can be easily satisfied by very small (near zero) initialization. According to the random matrix theory Theorem 2.7.5 in Tao [2012], for Gaussian initialization there exists $\nu > 0$ such that with high probability $\cos\boldsymbol{\theta}_0^u$ and $\cos\boldsymbol{\theta}_0^v$ is lower bounded by constant $1/\nu$, therefore one can simply set the norm of $\boldsymbol{U}_0$ and $\boldsymbol{V}_0$ to be sufficiently small such that the inequality (21) holds. In consequence small initialization can guarantee the global linear conver-

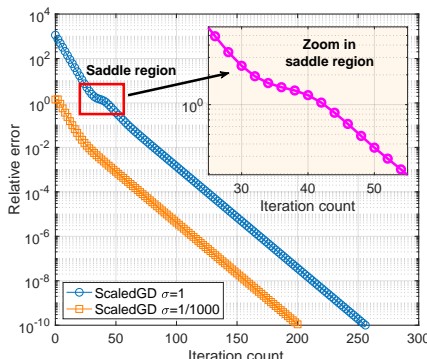

Figure 3: Global convergence of small initialization and general random initialization.

gence of ScaledGD, as shown in Fig. 3. While small initialization is very special. it can not helps us fully understand the global convergence property of ScaledGD from arbitrary initialization for the non-convex objective (1), even though small initialization has been widely used in the global convergence analysis of gradient descent algorithms Stöger and Soltanolkotabi [2021], Ye and Du [2021], Ma and Fattahi [2022] and ScaledGD for symmetric low rank matrix recovery problems Xu et al. [2023], Zhang et al. [2021].

### 4.1.2 General random initialization

In order to understand the optimization path of ScaledGD for the non-convex objective (1), we present the theoretical analysis of ScaledGD from random Gaussian initialization that may not satisfy the condition in Eq. (21). As shown in Fig. 3, when initialized with $\sigma = 1$ ScaledGD iterations are also attracted by the saddle point thus enter the saddle region (zoomed region marked by red rectangle), while it can escape saddle region very fast. To rigorously characterize the saddle avoid phase, we first show and prove that the norm of matrices $\boldsymbol{U}_k$ and $\boldsymbol{V}_k$ decrease if $\|\boldsymbol{M}\|_F \max\{\cos\boldsymbol{\theta}_k^u, \cos\boldsymbol{\theta}_k^v\} < \|\boldsymbol{U}_k\boldsymbol{V}_k^\top\|_F$ as given by the following lemma and shown in Fig. 4.

**Lemma 4.** *If the condition* $\|\boldsymbol{M}\|_F \max\{\cos\boldsymbol{\theta}_k^u, \cos\boldsymbol{\theta}_k^v\} < \|\boldsymbol{U}_k\boldsymbol{V}_k^\top\|_F$ *is satisfied then we have*

$$\|\boldsymbol{U}_{k+1}\|_F < \|\boldsymbol{U}_k\|_F \quad \text{and} \quad \|\boldsymbol{V}_{k+1}\|_F < \|\boldsymbol{V}_k\|_F. \tag{22}$$

*Furthermore, if the condition* $\|\boldsymbol{M}\|_F \cos\boldsymbol{\theta}_k^u \cos\boldsymbol{\theta}_k^v \geq \|\boldsymbol{U}_k\boldsymbol{V}_k^\top\|_F$ *is satisfied then we have*

$$\|\boldsymbol{U}_{k+1}\|_F \geq \|\boldsymbol{U}_k\|_F \quad \text{and} \quad \|\boldsymbol{V}_{k+1}\|_F \geq \|\boldsymbol{V}_k\|_F. \tag{23}$$

In general, if we initialize the matrices $\boldsymbol{U}$ and $\boldsymbol{V}$ as $\mathcal{N}(0, \sigma)$ with large $\sigma$, then in the initial phase, with high probability we have $\|\boldsymbol{M}\|_F \max\{\cos\boldsymbol{\theta}_0^u, \cos\boldsymbol{\theta}_0^v\} < \|\boldsymbol{U}_0\boldsymbol{V}_0^\top\|_F$. According to Lemma 3 and Lemma 4, we know that the norm of the matrices $\boldsymbol{U}_k$ and $\boldsymbol{V}_k$ decreases with the increase of $k$ until it reaches the condition $\|\boldsymbol{M}\|_F \cos\boldsymbol{\theta}_k^u \cos\boldsymbol{\theta}_k^v \geq \|\boldsymbol{U}_k\boldsymbol{V}_k^\top\|_F$, which is also illustrated in Fig. 4. In Fig. 4, we plot the changes of the norm of matrices $\boldsymbol{U}$ and $\boldsymbol{V}$, the nested subfigure illustrates the the matrix norm in log scale. It is very interesting to study the changes of the matrix norm with respect to the optimization path. Generally, if $\|\boldsymbol{U}_0\|_F$ and $\|\boldsymbol{V}_0\|_F$ is initialized very large, then the decrease of the norm will decrease the objective function (1). Meanwhile, $\boldsymbol{U} = \boldsymbol{0}$ and $\boldsymbol{V} = \boldsymbol{0}$ is a saddle point of the objective function (1), the results in Lemma 4 thus indicate that the matrices $\boldsymbol{U}_k$ and $\boldsymbol{V}_k$ are updated toward the saddle point zero. While interestingly, as shown in Fig. 4 the matrix norm decreases to a magnitude which is strictly larger than zero, then the matrix norm begins to increase. These observation indicates that ScaledGD can escape from the saddle point zero, the saddle avoid phase is also illustrated in Fig. 5.

**Analysis on the entire iteration process.** It can be easily deduced from Eq. (16) that

$$\|\boldsymbol{U}_k\boldsymbol{V}_k^\top - \boldsymbol{M}\|_F < (1-\eta)^{2k}\|\boldsymbol{U}_0\boldsymbol{V}_0^\top - \boldsymbol{M}\|_F + \|\boldsymbol{M}\|_F, \tag{24}$$

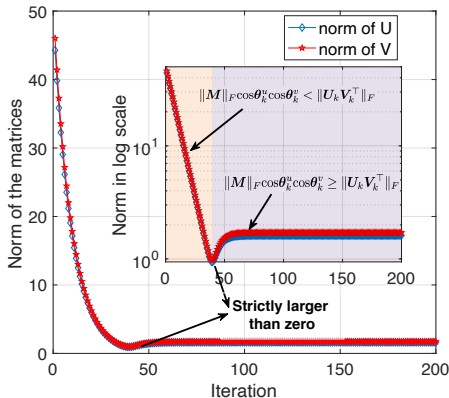

Figure 4: Illustration of the norms of matrices $\boldsymbol{U}$ and $\boldsymbol{V}$.

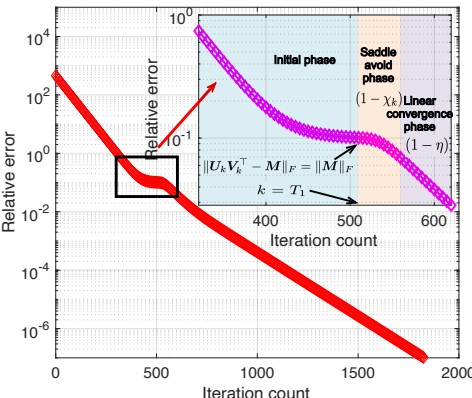

Figure 5: Illustration of the saddle avoid phase.

therefore after $T_1 = O(\ln\frac{1}{\delta})^3$ iterations (for sufficiently small $\delta$)[3], we have

$$\|\boldsymbol{U}_k\boldsymbol{V}_k^\top - \boldsymbol{M}\|_F \leq \|\boldsymbol{M}\|_F, \ \forall k \geq T_1, \tag{25}$$

which indicates that $\|\boldsymbol{M}\|_F\cos\boldsymbol{\theta}_k^u\cos\boldsymbol{\theta}_k^v \geq \frac{1}{2}\|\boldsymbol{U}_k\boldsymbol{V}_k^\top\|_F$. We term this period of time the **initial phase**. The following lemma tells that after $T_1$ iterations the term ②, ③ and ④ decrease linearly.

**Lemma 5.** *After $T_1$ iterations of ScaledGD, the following inequalities hold $\forall k \geq T_1$*

$$\|\mathcal{U}_{*\perp}^\top\mathcal{U}_{k+1}\|_2 \leq (1-\chi_k)\|\mathcal{U}_{*\perp}^\top\mathcal{U}_k\|_2 \tag{26}$$

$$\|\mathcal{V}_{*\perp}^\top\mathcal{V}_{k+1}\|_2 \leq (1-\chi_k)\|\mathcal{V}_{*\perp}^\top\mathcal{V}_k\|_2 \tag{27}$$

$$1 - \cos\boldsymbol{\theta}_{k+1}^u\cos\boldsymbol{\theta}_{k+1}^v \leq (1-\chi_k)^2\left(1 - \cos\boldsymbol{\theta}_k^u\cos\boldsymbol{\theta}_k^v\right) \tag{28}$$

*where $\chi_k = \frac{\eta\tau_k}{1-\eta(1-\tau_k)} < 1$ and $\tau_k = \frac{\|\boldsymbol{M}\|_F\cos\boldsymbol{\theta}_k^u\cos\boldsymbol{\theta}_k^v}{\|\boldsymbol{U}_k\boldsymbol{V}_k^\top\|_F} \in [1/2, 1]$.*

If $\frac{1}{2}\|\boldsymbol{U}_k\boldsymbol{V}_k^\top\|_F \leq \|\boldsymbol{M}\|_F\cos\boldsymbol{\theta}_k^u\cos\boldsymbol{\theta}_k^v \leq \|\boldsymbol{U}_k\boldsymbol{V}_k^\top\|_F$ and $\|\boldsymbol{M}\|_F \geq \|\boldsymbol{U}_k\boldsymbol{V}_k^\top\|_F$, we have that the term ④ in Eq. (16) is upper bounded by $1 - \cos\boldsymbol{\theta}_k^u\cos\boldsymbol{\theta}_k^v$. Thus the above Lemma 5 indicates that after $T_1$ iterations, the objective function decreases at rate $1 - \chi_k$. Meanwhile, $\cos\boldsymbol{\theta}_k^u$ and $\cos\boldsymbol{\theta}_k^v$ are increasing, and $\|\boldsymbol{U}_k\boldsymbol{V}_k^\top\|_F$ continues to decrease until $\|\boldsymbol{M}\|_F\cos\boldsymbol{\theta}_k^u\cos\boldsymbol{\theta}_k^v \geq \|\boldsymbol{U}_k\boldsymbol{V}_k^\top\|_F$, which means the value $\tau_k = \frac{\|\boldsymbol{M}\|_F\cos\boldsymbol{\theta}_k^u\cos\boldsymbol{\theta}_k^v}{\|\boldsymbol{U}_k\boldsymbol{V}_k^\top\|_F}$ is monotonically increasing with the increase of $k$ until up to 1. In consequence, the $\chi_k$ is monotonically increasing from $\frac{\eta/2}{1-\eta/2}$ to $\eta$. We name the period in which $\chi_k$ increases from $\frac{\eta/2}{1-\eta/2}$ to $\eta$ the **saddle avoid phase** as shown in Fig. 5 [4]. The Lemma 5 also indicates that the ScaledGD escapes saddle points exponentially fast. After the saddle avoid phase, the ScaledGD converges to the global minima at rate $1 - \eta$ according to the analysis in Section 4.1.1, since $\|\boldsymbol{M}\|_F\cos\boldsymbol{\theta}_k^u\cos\boldsymbol{\theta}_k^v \geq \|\boldsymbol{U}_k\boldsymbol{V}_k^\top\|_F$, we name this period the **linear convergence phase** as shown in Fig. 5.

### 4.2 Convergence of AltScaledGD

The convergence analysis of the AltScaledGD is similar to that of ScaledGD, while different to ScaledGD, the objective function (1) is upper-bounded by three terms in AltScaledGD as

$$\|\boldsymbol{U}_{k+1}\boldsymbol{V}_{k+1}^\top - \boldsymbol{M}\|_F \leq \underbrace{(1-\eta)^2\|\boldsymbol{U}_k\boldsymbol{V}_k^\top - \boldsymbol{M}\|_F}_{①} + \underbrace{(\eta-\eta^2)\|\boldsymbol{M}\|_F\|\mathcal{V}_{*\perp}^\top\mathcal{V}_k\|_2}_{②} + \underbrace{\eta\|\boldsymbol{M}\|_F\|\mathcal{U}_{*\perp}^\top\mathcal{U}_{k+1}\|_2}_{③}$$

$$\tag{29}$$

---

[3]Please refer to the supplementary results for more detailed analysis.

[4]Since in this period of time, the norm of the matrices $\boldsymbol{U}_k$ and $\boldsymbol{V}_k$ decrease, while once $\chi_k = \eta$ (equivalently $\tau_k = 1$), according to Lemma 4 and Lemma 3, the norm of the matrices $\boldsymbol{U}_k$ and $\boldsymbol{V}_k$ begin to increase, which indicates that the matrices $\boldsymbol{U}_k$ and $\boldsymbol{V}_k$ are escaping from the saddle point zero.

therefore, the analysis of AltScaledGD for (1) is much easier than that of the ScaledGD in Eq. (16). Specifically, we only need to guarantee that the distance between subspaces of $\boldsymbol{U}_k$ and $\boldsymbol{U}_*$ ($\|\mathcal{U}_{*\perp}^\top \mathcal{U}_k\|_2$), $\boldsymbol{V}_k$ and $\boldsymbol{V}_*$ ($\|\mathcal{V}_{*\perp}^\top \mathcal{V}_k\|_2$) decrease linearly. The Lemma 2 also holds for AltScaledGD as

**Lemma 6.** (Convergence of the distance between subspaces) *For AltScaledGD (4), if* $\|\boldsymbol{M}\|_F \cos\boldsymbol{\theta}_k^u \cos\boldsymbol{\theta}_k^v \geq \|\boldsymbol{U}_k \boldsymbol{V}_k^\top\|_F$ *and* $0 < \eta \leq 1$, *then the following holds*

$$\|\mathcal{U}_{*\perp}^\top \mathcal{U}_{k+1}\|_2 \leq (1-\eta)\|\mathcal{U}_{*\perp}^\top \mathcal{U}_k\|_2, \quad \|\mathcal{V}_{*\perp}^\top \mathcal{V}_{k+1}\|_2 \leq (1-\eta)\|\mathcal{V}_{*\perp}^\top \mathcal{V}_k\|_2. \tag{30}$$

The condition in Lemma 6 can be satisfied $\forall k$ if $\|\boldsymbol{M}\|_F \cos\boldsymbol{\theta}_0^u \cos\boldsymbol{\theta}_0^v \geq \|\boldsymbol{U}_0 \boldsymbol{V}_0^\top\|_F$ and $0 < \eta \leq 1$ as specified by the following lemma, the condition is mild compared to the condition in Lemma 3.

**Lemma 7.** *For AltScaledGD (4), if* $\|\boldsymbol{M}\|_F \cos\boldsymbol{\theta}_0^u \cos\boldsymbol{\theta}_0^v \geq \|\boldsymbol{U}_0 \boldsymbol{V}_0^\top\|_F$ *and* $0 < \eta \leq 1$, *then the following is true*

$$\|\boldsymbol{M}\|_F \cos\boldsymbol{\theta}_k^u \cos\boldsymbol{\theta}_k^v \geq \|\boldsymbol{U}_k \boldsymbol{V}_k^\top\|_F, \forall k > 0. \tag{31}$$

The convergence analysis of AltScaledGD is the same as that of the ScaledGD in Section 4.1 with small initialization and general Gaussian initialization (the three phases convergence). The main difference between ScaledGD and AltScaledGD is that $\eta$ in ScaledGD should be small such that $\eta \leq c_\eta < 1$, while for AltScaledGD $\eta$ can be as large as 1. It can be seen from Eq. (29) that if $\eta = 1$, the the AltScaledGD Eq. (4) converges to the global minima in just one iteration. We also illustrate the convergence of ScaledGD Eq. (2) and AltScaledGD Eq. (4) with respect to the learning rate $\eta$ in Fig. 6. It can be seen from Fig. 6 that for small learning rate $\eta = 0.1$, the convergence property (the loss curve) of AltScaledGD is almost exactly the same as ScaledGD, while for large learning rate $\eta = 0.8$, the

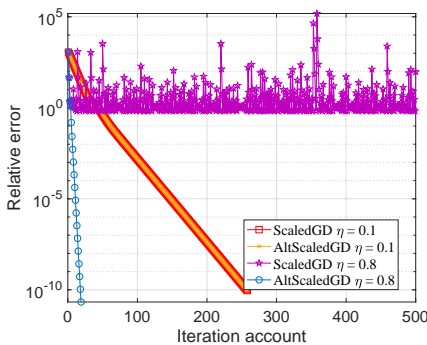

Figure 6: Illustration of the effect of learning rate $\eta$ for the convergence.

AltScaledGD converges very fast, in contrast the ScaledGD does not converge as the condition $\eta \leq c_\eta$ is not satisfied according to Lemma 3. These results certificates the superiority of AltScaledGD over ScaledGD, since both ScaledGD and AltScaledGD converges fast with large $\eta$, while the learning rate $\eta$ is upper-bounded by a small constant $c_\eta$ in ScaledGD.

## 5  Conclusion

In this work, we are the first to rigorously prove the global convergence of ScaledGD and AltScaledGD for the non-convex low rank matrix factorization problem and show that thanks to the preconditioning matrices the global convergence rate of ScaledGD and AltScaledGD are independent of the condition number of the matrix $\boldsymbol{M}$, thus they converge faster than gradient descent algorithm for ill-conditioned problem. We further prove that ScaledGD and AltScaledGD converges linearly from both small initialization and general random initialization, which is in contrast to the existing global convergence analysis that are only applicable to small initialization. Meanwhile, we show that compared to ScaledGD, AltScaledGD is more practical as it enables larger learning rate thus converges fast.

**Limitations.** This paper concerns low-rank matrix factorization which is the population loss of the more general low-rank matrix recovery problem, such as matrix completion and matrix sensing. While the empirical loss is different to the population loss in that the number of the samples is limited, therefore our results can not directly applied to general low-rank matrix recovery. Our further work is to study the empirical loss with the help of RIP condition for matrix sensing and the sampling lower-bound for matrix completion.

### Acknowledgements

This research was supported by the National Key R&D Program of China (2020YFA0713900), the China NSFC projects under contracts 62372359, 61721002, 12226004, the Macao Science and Technology Development Fund under Grant 061/2020/A2, and the Fundamental Research Funds for the Central University under Grant ZYTS23056.

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
