# OpenReview forum: "Preconditioning Matters: Fast Global Convergence of Non-convex Matrix Factorization via Scaled Gradient Descent"
_NeurIPS.cc/2023/Conference — NeurIPS 2023 poster_

### Official Review · Reviewer_Cp2y · 2023-07-03

**Soundness:** 3 good
**Presentation:** 3 good
**Contribution:** 3 good
**Rating:** 6
**Confidence:** 4

**Summary:**

This paper considers the low-rank matrix factorization problem (LRMF).  Recent work provided global convergence for gradient descent on LRMF starting from small random initialization and small learning rate, but the convergence rate there depends on the matrix condition number.  This paper considers a variant of gradient descent where the update is rescaled akin to steepest descent, called ScaledGD (and its alternating version, AltScaledGD). The main result of this paper is proving that ScaledGD (and AltScaledGD) converges at a faster rate, with no dependence on the matrix condition number.  Moreover, AltScaledGD converges even without small initialization and small step-size.

**Strengths:**

The main theoretical results of this paper are quite solid.  ScaledGD (and AltScaledGD) are very reasonable algorithms for solving the LRMF problem, and this paper proves that they converge at a fundamentally faster rate than plain GD on the LRMF problem.







**Weaknesses:**

The presentation is pretty good, although it does seem written in a rush in many places.

The main weakness of this paper is its motivation.
- The low-rank matrix factorization problem can solved very efficiently using specialized matrix algorithms, so if one studies simple general algorithms like gradient descent for LRMF, the end goal should not be to use it for LRMF, but rather to gain some understanding on the behavior of the simple algorithm on more complicated optimizations like neural networks.  But while gradient descent is feasible on more complicated problems, I do not see how to implement ScaledGD (or AltScaledGD) on neural network optimization problems, or on any optimization problem where a specialized algorithm can already do better.

**Questions:**

To address the concerns in the 'weaknesses' box, please defend the motivation for studying ScaledGD (or AltScaledGD).  Could these algorithms reasonably be implemented on nonlinear neural network optimization problems (or on optimization problems beyond matrix factorization where it could be the state-of-art algorithm)?


Can you provide experimental evidence that the algorithms here perform significantly better than GD (or AltGD) when the underlying matrix is not exactly low-rank, but just approximately low-rank?  Of course, any algorithm for LRMF should remain competitive when the matrix M is not exactly low-rank.







**Limitations:**

yes

---

> ### Author Rebuttal · Authors · 2023-08-06
>
> We greatly appreciate the reviewer for your positive evaluation of our work, and we also thank you for your valuable and constructive suggestions. As for your concerns, we make detailed responses as follows. We will deeply appreciate that you can raise your score if you find our responses resolve your concerns.
>
> **Q1: The motivation for studying ScaledGD/AltScaledGD on LRMF and could these algorithms be implemented on nonlinear neural network optimization problems?**
> **A1**: Gradient descent algorithm is of great significance in non-convex optimization problems as studied by recent works Chi et al. [2019], Li et al. [2019a], Ye and Du [2021]. ScaledGD is a simple extension of GD by scaling the gradient with a precondition matrix with minimal computational overhead, while it resolves the convergence barrier of GD for ill-conditioned non-convex optimization problems such as low rank matrix recovery (LRMR) Tong et al. [2021]. ScaledGD is also computationally more efficient than the current SOTA algorithm for the LRMR problem as analyzed in Tong et al. [2021].
>
> While the current convergence analysis of ScaledGD for LRMR in Tong et al. [2021] is only local (with spectral initialization), it is more important to study the global convergence of the ScaledGD for the non-convex LRMR problem. Our work makes the first attempt to study the global convergence of ScaledGD and AltScaledGD for asymmetric LRMF, and our results can be directly extended to general low rank matrix recovery problem with the help of conditions such as RIP for matrix sensing.
>
> The ultimate goal of studying ScaledGD is to use it for more non-convex optimization problems such as the training of deep neural networks. We show our recent results of ScaledGD and AltScaledGD on more challenged optimization problems such as deep linear network and nonlinear network in Figure 1 and Figure 2 of the PDF file in the Author rebuttal part.
>
> Specifically, we consider the following deep linear model:
> $$\min_{W_3, W_2, W_1} f(W_1, W_2, W_3) := \frac{1}{2} ||W_3W_2W_1 - M||_F^2$$
>
> with the ScaledGD iteration given by:
>
> $$W_1^{k+1} = W_1^{k} - \eta (W_2^{k\top} W_3^{k\top}W_3^k W_2^k)^{-1} \textcolor{red}{{{\nabla} _{W_1}}f}$$
>
> $$ W_2^{k+1} = W_2^{k} - \eta (W_3^{k\top} W_3^k)^{-1} \textcolor{red}{{{\nabla} _{W_2}}f} (W_1^kW_1^{k\top})^{-1}$$
>
> $$W_3^{k+1} = W_3^{k} - \eta  \textcolor{red}{{{\nabla} _{W_3}}f}(W_2^k W_1^kW_1^{k\top} W_2^{k\top})^{-1} $$
>
>  and the nonlinear model:
> $$\min_{W_2, W_1} f(W_1, W_2) := \frac{1}{2} ||W_2\sigma(W_1) - M||_F^2$$
>
> with ScaledGD iteration given by:
>
> $$W_1^{k+1} = W_1^k - \eta {G^k} \odot \left( (W_2^{k\top} W_2^k)^{-1} (H^k\odot  \textcolor{red}{{{\nabla} _{W_1}}f})\right)$$
>
> $$W_2^{k+1} = W_2^k - \eta  \textcolor{red}{{{\nabla} _{W_2}}f }\left( \sigma(W_1^k) \sigma(W_1^k)^{\top}\right)^{-1}$$
>
> where $G^k$ is a matrix with $G_{i,j}^k = \frac{\partial \sigma(W_{1_{i,j}}^k)}{\partial W_{1_{i,j}}^k}$, $H^k$ is a matrix with elements $H_{i,j}^k = 1/G_{i,j}^k$, $\eta$ is the step-size and $\sigma(\cdot)$ is a piece-wise linear function as Leaky ReLU. It can be seen from Figure 1 that ScaledGD converges much faster than GD and the convergence rate is independent of the condition number $\kappa$ of the target matrix $M$, whereas GD convergence at different rates for different $\kappa$. Figure 2 shows the result of ScaledGD/AltScaledGD for the nonlinear network problem. Figure 2 (a) plots the loss curves of ScaledGD and GD under different condition number $\kappa$, Figure 2 (b) plots the zoomed-in curves of GD in Figure 2 (a). It can be seen from these figures that ScaledGD converges much faster than GD even for this nonlinear network model, and the convergence rate of ScaledGD is also independent of the $\kappa$, while in contrast GD converges at different rate under different $\kappa$. In Figure 2 (c), we compare ScaledGD with AltScaledGD, it can be seen that both ScaledGD and AltScaledGD converges linearly to the global minima, while AltScaledGD converges faster than ScaledGD. Please refer to the PDF file for the figures.
>
> **Q2: Provide experimental evidence to show that the algorithms here perform significantly better than GD when the underlying matrix is not exactly low-rank.**
> **A2**: We have tested the proposed methods on real data sets such MSI data and video data which are not exactly low-rank but are approximately low-rank. We show the experimental results in Figure 3 and Figure 4 in the attached PDF file at the author rebuttal part. It can be seen from these figures that even for real data sets with not exactly low-rank, the ScaledGD and AltScaledGD still converge much faster than the vanilla GD.

---

> ### Author Response · Authors · 2023-08-17
> **Request for discussion**
>
> Dear reviewer Cp2y,
>
> We understand that reviewing is a time-consuming task and we want to express our gratitude for your dedication. Since the end of the discussion period is approaching, if you have any further concerns please feel free to let us know and we are pleased to discuss with you. We will greatly appreciate that you can raise your score if you find our responses resolve your concerns.
>
> Thank you very much !

---

> ### Author Response · Authors · 2023-08-20
>
> Dear reviewer,
>
> Since the end of the discussion period is approaching, if you have any further concerns please feel free to let us know and we are pleased to discuss with you. Thank you very much!

---

> > ### Comment · Reviewer_Cp2y · 2023-08-21
> >
> > I have read all the comments, and I will maintain my score.

---

### Official Review · Reviewer_beQc · 2023-07-03

**Soundness:** 3 good
**Presentation:** 1 poor
**Contribution:** 3 good
**Rating:** 4
**Confidence:** 4

**Summary:**

This paper considers the problem of low-rank matrix recovery using preconditioned gradient descent. Traditionally, although GD can be used to solve this problem, it becomes extremely slow when the problem is ill-conditioned. Recently a method called ScaledGD was proposed that makes GD immune to ill-conditioning. However, previously, the theoretical analysis of ScaledGD required spectral initialization, and this work extends that analysis to small or moderate random initialization. The authors also analyze an alternating version of ScaledGD, which they call AltScaledGD, and also prove global convergence for this algorithm.

**Strengths:**

Previously, algorithms like ScaledGD with make GD immune to ill-conditioning relies on an initial point close to the ground truth. Although this can be achieved through a method known as spectral initialization, it can be very expensive. This work strengthens previous results by showing that a *random* initialization can also achieve linear convergence. Similar results have been established for vanilla GD with small initialization. Here the authors extend such results to ScaledGD, and show that a moderate random initialization also works. To me this is the main novelty of this work.

**Weaknesses:**

The main weakness that prevents me from giving this work a higher score is the organization of the technical sections. In the introduction the authors claim that they prove global convergence of ScaledGD with moderate initialization. However, in section 3.1 the main results are only stated for the **rank-1** case. Again, in section 4, the authors claim that they will present the theoretical analysis for the rank-1 case. In the appendix, however, they authors present Theorem 3, which is the moderate initialization case for the rank-d case. But the proof of Theorem 3 refers to some results in section 4, which was presented for the rank-1 case. As a result, it is hard to decipher which parts of the rank-1 case generalize directly to the rank-d case, and which parts need more work.

Overall I think the organization of the technical sections is confusing and needs significant improvement.






**Questions:**

N/A

---

> ### Author Rebuttal · Authors · 2023-08-07
>
> We would like to express our sincere gratitude for your valuable comments on our work. As for your concerns, we give detailed response below which we hope can help you fully understand our work. Please feel free to let us know if you have any further concerns. We will deeply appreciate that you can raise your score if you find our responses resolve your concerns.
>
> **Q1: The organization of the technical sections should be improved, some of the results are only stated for rank-1 and the proof from rank-1 to rand-d is confusing.**
> **A1**: The overall picture of the technical sections is that we first present the main results (rank-1 for ScaledGD and rank-d for AltScaledGD) in section 3.  Then to have a proof guideline, we give the proof sketch of the rank-1 case of both ScaledGD and AltScaledGD in section 4, as the rank-1 case is easy to understand and follow. At last, in Appendix we present more detailed proofs of both rank-1 and rank-d based on the proof sketch in section 4. Since the proof of rand-d follows the proof sketch of rank-1 with some moderate changes of the lemmas and inequalities from rank-1 to rank-d, we refer the proof sketch of rank-1 as the proof guideline of Theorem 3 in Appendix.  We are sorry to make you feel confused on our proof.
>
> Note that ScaledGD is closely related to GD but it has quite different convergence property compared to GD, if $rank(M)=1$, ScaledGD becomes exactly GD but with varying step-size. Therefore, to make a concise and clear comparison between ScaledGD with GD, we present the convergence results of ScaledGD for rank-1 case (the preconditioning relates to a number rather than a matrix). Then for AltScaledGD, to fully understand the convergence property we present the results for general rank-d in section 3.2, therefore section 3 consists of all our main results.
>
> Once again, we thank the reviewer for your valuable comments, we will carefully revise the current manuscript according to the suggestions provided by the reviewer to further enhance the writing.

---

> ### Author Response · Authors · 2023-08-17
> **Request for a discussion**
>
> Dear reviewer beQc,
>
> We understand that reviewing is a time-consuming task and we want to express our gratitude for your dedication. We value your expertise and opinion, and hope that you can take time to have discussions with us if you have any further concerns. We would greatly appreciate it if you could raise your score if our responses have addressed your concerns.
>
> Thank you very much !

---

> > ### Comment · Reviewer_beQc · 2023-08-19
> >
> > I thank the authors for the clarifications. However, I feel this organization is very confusing, even after the clarification. If I understand correctly: the main results of this paper are: rank-1 convergence for ScaledGD and rank-d convergence for AltScaledGD? In other words, the authors do not prove global convergence for rank-d ScaledGD, is this correct?

---

> > > ### Author Response · Authors · 2023-08-19
> > >
> > > Thank you for your response. To study the results of an optimization algorithm for rank-1 can help us gain more insight and understanding on the theoretical analysis, which has been widely used in low rank matrix recovery. We do not provide the results of ScaledGD for rank-d but instead provide the results of AltScaledGD since the results of ScaledGD follow directly from Theorem 1 and Theorem 2, and the corresponding proof is only trivial given all the results available (please refer to our discusses below).
> > >
> > > **1.	Why do we provide the convergence results for rank-1?**
> > > ScaledGD is closely related to GD and specifically it becomes exactly GD but with varying step-sizes (implemented by the scaling number) when $rank (M) = 1$. Since the comparisons of ScaledGD and GD in the rank-1 case is easy to understand and follow, we first presented the convergence results of ScaledGD for rank-1, which is similar to [1] to deal with the global convergence of GD.
> > >
> > > [1] Simon S Du, Wei Hu, and Jason D Lee. Algorithmic regularization in learning deep homogeneous models: layers are automatically balanced. Advances in Neural Information Processing Systems, 31:382–393, 2018.
> > >
> > > **2.	Why is the proof of ScaledGD for rank-d trivial?**
> > > We have provided detailed comparisons and analysis (the proof sketch) of the convergence of ScaledGD and AltScaledGD of rank-1 (section 4) for easier understanding. Based on the proof sketch of rank-1, the proof of the convergence of AltscaledGD for rank-d follows easily
> > > by extending the inequalities and lemmas from rank-1 to rank-d. With all these in hand, to prove the results of ScaledGD for rank-d is only trivial by following the proof guideline of its rank-1 case and the corresponding lemmas and results of the proof of AltScaledGD.
> > >
> > > If the reviewer concerns that the results provided in section 3.1 (for rank-1) make the organization of the paper confusing, we can simply replace the results in Theorem 1 and Theorem 2 to rank-d (our results for rank-1 can be directly generalized to rank-d) and append the aforementioned proof which only needs minor revision of the current Appendix.
> > >
> > > Once again, thanks for your comments and response.

---

> > > ### Author Response · Authors · 2023-08-19
> > >
> > > Dear reviewer,
> > >
> > > By far, your main concern is that the results provided in section 3.1 is for rank-1 rather than for rank-d. Yet, the overall picture and organization of this paper are clear. Meanwhile, to present the results for rank-1 is commonly used in low-rank matrix recovery. In our work, given the rank-1 results/proof of ScaledGD and the rank-d results/proof of AltScaledGD, the proof of ScaledGD for rank-d, which we omitted in the initial submission, is only trivial by using existing lemmas and inequalities which follows the proof sketch. According to the reviewer’s comment, we would like to revise the corresponding part in 3.1, **which only needs minor revision of the current manuscript.**
> > >
> > > We hope that replacing the results of Theorem 1 and Theorem 2 by rank-d as given by AltScaledGD can address your concern. Thank you very much.

---

> > > > ### Comment · Reviewer_beQc · 2023-08-19
> > > >
> > > > Thank you again for the clarifications. I will maintain my score for now.

---

### Official Review · Reviewer_eEnz · 2023-07-04

**Soundness:** 3 good
**Presentation:** 2 fair
**Contribution:** 2 fair
**Rating:** 6
**Confidence:** 2

**Summary:**

This paper studies  low rank matrix factorization, which is an important topic with many applications in machine learning. The main challenge of this problem comes from the non-convexity of the objective function, especially, this objective function can be non-smooth. As a consequence, the global convergence remains a difficult question in this area. The most recent approximate global minima convergence guarantee depends on quite a few parameters, and this, especially small initialization and small learning rate, might reduce the practical application potential of the convergence results. To address this issue, this paper shows that precondition helps in accelerating the convergence and the scaled gradient descent converges to approximate minima after a better number of iterations.

**Strengths:**

The problem in this paper is well motivated and studied. The main strength of this paper is the convergence result does not sensitively depend on condition number and small learning rate, which improve the application potential of the current algorithm and convergence analysis.

Another strength is that this paper provide some variants of the main algorithms and simultaneously provide comparable convergence rates, which might of independent interests.


**Weaknesses:**

1. The main weakness of this work is lacking of real data based experiments. Despite the theoretical analysis is complete and technical, matrix factorization is a powerful method in real application.

2. The convergence rate is theoretically improved, however, the practical impact of matrix factorization algorithm is inevitably tied with real data set experiments. It is not clear how the algorithm works in real data set, so it might cause some difficulty to judge the contribution of this theoretically sound paper.



**Questions:**

1. It is not quite obvious to me that why the initialization can be completely random. Since the objective function is non-convex, the initial points can still converge to different local minimum, and this might cause a difference in the resulting output.

2. Why the saddle avoidance phase hold for random initialization? I might miss some point, but one of the contribution of this paper is about the initialization can be random, and then, at least intuitively, this will cause some technical challenges in analyzing the saddle avoidance phase. Could you please provide some intuition on the proof of 157-176 in Appendix, so that it is more clear that the saddle avoidance can be done for random initialization?

3. Another question on the theoretical side is: what is the main technical challenge in proving the convergence of alternating scaled GD, compared to the main algorithm?

4. Line 71, why near zero initialization is an issue? How does it effect specific ill-conditioned LRMF problems?

5. Line 78, results of Tong et al are local or global? The presentation causes confusion.


**Limitations:**

As stated before, the main limitation of this paper is the experiment, it is not convincing that current theoretical results is powerful enough to have strong impact in applications.

---

> ### Author Rebuttal · Authors · 2023-08-07
>
> We greatly thank the reviewer for the valuable comments and the constructive suggestions. Here we response your concerns one by one. We will deeply appreciate that you can raise your score if you find our responses resolve your concerns.
>
> **Q1: How does the algorithms work on real data set.**
> **A1**: The ScaledGD and AltScaledGD are not only work on simulated data but also on real data set such as multi-spectral image (MSI) data and video data sets. We provide a PDF file in the author rebuttal part, and report the experimental results of GD, ScaledGD and AltScaledGD on MSI and video data sets in Figure 3 and Figure 4. In these figures, we can see that ScaledGD and AltScaledGD converges much faster than GD on both MSI data sets and video data sets. Specifically, the relative error of ScaledGD/AltScaledGD is less than $10^{-2}$ for MSI data "flower" and is less than $10^{-5}$ for MSI data "Simu_Indian" within $500$ iterations, while the relative errors of GD are still very large after $2000$ iterations. The experimental results are consistent with the theoretical results provided in the main paper. We will also replenish these real data sets experiments on the Appendix of our revision.
>
> Meanwhile, our follow-up works indicate that ScaledGD/AltScaledGD work well for more applications, please refer to our response to Q1 of reviewer Cp2y and Figure 1,  Figure 2 on the PDF file which show that ScaledGD/AltScaledGD work well for neural network model.
>
> Overall, this conference paper focuses on proving the global convergence of ScaledGD and AltScaledGD for the non-convex LRMF model and analyze the descent trajectory of the algorithms. The results are significant improvements to the work of Tong et al. 2021 for the analysis of the ScaledGD. More applications of ScaledGD and AltScaledGD and their convergence analysis are our ongoing work.
>
> **Q2:  Why the initialization can be completely random.**
> **A2**: For non-convex optimization, if the objective functions satisfy the strict saddle property (SSP) that is all saddle points have a decent direction, then gradient based algorithm can converge to local minima [1]. The low rank matrix factorization as well as low rank matrix recovery satisfy the SSP, meanwhile, the objective functions have benign landscape (all local minima are global minima) [2], therefore even with random initialization ScaledGD/AltScaledGD is able to converge to the global minima. While, existing works do not provide global convergence rate of ScaledGD/AltScaledGD on asymmetric LRMF, we are the first to give a detailed convergence rate analysis of ScaledGD and AltScaledGD with random initialization.
>
> [1] Lee, J. D., et al. Gradient descent only converges to minimizers. In COLT, 2016.
> [2] Ge, R., et al. Matrix completion has no spurious local minimum. In NeurIPS, 2016.
>
> **Q3: Provide some intuition on the proof of 157-176 for the saddle avoid.**
> **A3**: The saddle avoid phase (line 157-176 in Appendix) is analog to the rank-1 case analyzed in section 4.1.2 line 295-305 in the main paper, which guarantees that ScaledGD does not trap into the saddle point. Specifically, for general Gaussian initialization with the scale of the initial valuable $U_0$ and $V_0$ greater than some constant $c_{init}$, according to Lemma 4, we know that the norm of the matrix $U$ and $V$ decrease toward the saddle point $U = 0$ and $V = 0$ until the inequality $ <U_k V_k^{\top}, M > \geq || U_k  V_k^{\top}||_F^2$ is satisfied. The inequality is crucial to our analysis since once it is satisfied, according to Lemma 4 the norm of the matrix $U_k V_k^{\top}$ tends to increase as shown in Figure 4 of the main paper, and meanwhile Lemma 5 and Lemma 6 in the main paper show that the angle between the subspace of $U_k$ ($V_k$) and $ U^*$ ($ V^*$) decrease linearly, which reveals that the ScaledGD has escaped the saddle region (since in rank-1 case the saddle point is 0 matrix with norm 0).
>
> The proof of 157-176 in Appendix is to show that after the initial phase we have $< U_kV_k^{\top}, M> \geq \tau_k ||U_kV_k^{\top}||_F^2$, and in the saddle avoid phase the variable $\tau_k$ is increasing from 1/2 to 1 such that the inequality $<U_k V_k^{\top}, M> \geq ||U_kV_k^{\top}||_F^2$ is fulfilled. After $\tau_k$ grows up to 1, the algorithm enters the linear convergence phase as analyzed in line 177-185 in Appendix.
>
> **Q4: Technical challenge in proving the convergence of alternating scaled GD ?**
> **A4**: The proof of the convergence of the AltScaledGD is similar to that of the ScaledGD, with only difference in that the term ④ in Eq. (12) limits the learning rate $\eta$ to be smaller than a constant $c_{\eta}$, while there is no corresponding term ④ in Eq. (25), therefore the learning rate $\eta$ can be set as $0\leq \eta \leq 1$ in AltScaledGD. In consequence, AltScaledGD enjoys lager learning rate and faster convergence rate than ScaledGD as specified in line 318-336 of the main paper.
>
> **Q5: Why near zero initialization is an issue?**
> **A5**: Near zero initialization as well as spectral initialization limit the initialization to a local region near some certain points, which has strong initialization bias. While for the theoretical analysis of non-convex optimization, one is interested in the global convergence property of the optimization algorithm which starts from any random initialization (without initialization bias) rather than a local region of some certain point. We prove that the ScaledGD and AltScaledGD are not sensitive to the scale of the initialization, they converge fast in both general random initialization and near zero initialization.
>
> **Q6: Results of Tong et al. are local or global?**
> **A6**: The results of Tong et al. are local, since it relies on the spectral initialization (initialization in a local region the global minima). In contrast, our results are global as our initialization is random without initialization bias. We will revise our statement in the main paper.

---

> > ### Comment · Reviewer_eEnz · 2023-08-14
> >
> > Thanks for the additional experiments and answers to my questions, I have no questions at this point. The score can be improved due to my concerns on real data experiments has been addressed.

---

> > > ### Author Response · Authors · 2023-08-15
> > > **Acknowledgment**
> > >
> > > We sincerely appreciate your time and efforts in providing us with your response. Your support has made a significant difference in our work, and we are confident that it will lead to a stronger final product.

---

### Official Review · Reviewer_QnG3 · 2023-07-26

**Soundness:** 3 good
**Presentation:** 3 good
**Contribution:** 3 good
**Rating:** 7
**Confidence:** 2

**Summary:**

This paper considers the classical problem of low rank approximation. In particular, given a matrix $M\in \Re^{m \times n}$ with rank $d$, we want to find $U\in \Re^{m\times d}, V\in \Re^{n\times d}$ that minimizes $f(U,V) = |UV^T - M|_F$ (i.e $U,V$ are low rank matrices whose product approximate $M$). The problem is very well studied and classical.

Obviously, one can perform classical gradient descent on $U,V$ to optimize for $f$ (although this is not desirable because it relies on the condition number of $M$). It is shown that the local minima of $f$ do not misbehave, so gradient descent works fine. However, in the literature, there are two "gradient scaled" iterative variants of gradient descent that "scale" the gradient by right multiplying it by appropriate matrices. This is well studied in the literature, and the two variants considered in the paper in ScaledGD and AltScaledGD (where AltScaledGD is similar to ScaledGD but "alternates" the scaling matrix multiplied by the gradient). There has been a lot of work that studies the theoretical convergence of these two problems. The latest is the work from 2021 by Ye and Du that get linear convergence until very strong assumptions (technically, they're not "assumptions", but the convergence rate is not clean at all).

In this paper, the authors give an elegant proof that both ScaledGD and AltScaledGD converge linearly for the low rank approximation problem. This is the first "clean" analysis (in the sense that all the constants are clear) for the two algorithms.

**Strengths:**

- The paper is cleanly written, and reads well.
- The analysis is quite elegant and sheds a lot of answers on the intuition on why these algorithms work well and do not rely on the condition number of $M$.
- Some of the lemmas proven along the way might me of independent interest (for example, while I am not related directly to this line of research, one of the Lemmas proven will probably very useful for my next paper). The proofs are well written (although there are some rough patches there).

**Weaknesses:**

- Some areas are rushed in the main paper. Some areas need to be explicitly written instead of using "it can be deduced" which is frustrating to the reader and is used a couple of times by the authors (when it really shouldn't). I've noted some examples below, but please avoid doing this and write the steps explicitly. I understand this would take longer to write, but it makes the reading process much more pleasant.
- The author(s) can spend a bit more time on explaining the intuition behind some of the inequalities they derive. The proofs sometimes seem very mechanical, and in other areas seem to shed a lot of intuition on what is happening. It would be nice if this is standardised so that the overall picture of the proof is explained first, before churning out the math. For example, some of the proofs from Section 3 are quite heavy and go through math without offering why said computations are being made. It's only in the end after rereading that you get what was happening.

------------------------------------------------------------------
Lines 80: AltScaledGD is not introduced earlier. Please introduce it earlier.

Line 108: Scaled -> scaling

Line 117: "converges much faster than gradient descent while ther" ==> "converges much faster than gradient descent. However, ...."

Line 120-122: "The local convergence .. problem (1)." This is an unnecessary sentence. Remove or rephrase.

Line 154: Theorem 1, I might be missing something, but where is the dependence on \sigma in Theorem 1 statement?

Line 216: "It can be deduced tha.." Don't do that, please spell it out explicitly (i.e sub in eq(2) and expand..


**Questions:**

Line 154: Theorem 1, I might be missing something, but where is the dependence on \sigma in Theorem 1 statement?

**Limitations:**

Lines 346-351. Well addressed.

---

> ### Author Rebuttal · Authors · 2023-08-06
>
> We deeply appreciate the reviewer for your careful review, constructive suggestions and positive feedbacks. The followings are our responses to your concerns. We will greatly appreciate that you can raise your score if you find our responses resolve your concerns.
>
> **Q1: Write some of the proof steps explicitly.**
> **A1**: We will carefully check our proof and rewrite the proof with more details such that it is easy to read. Thank you for your kind suggestions.
>
> **Q2: Explain the overall picture of the proof and the intuition behind some of the inequalities.**
> **A2**: To help us understanding the proof of the main results in section 3, we provided the proof sketch of the main results for rank-1 case in section 4 since it is easy to follow and understand (the proof sketch of AltScaledGD for rank-1 case is the similar to that of ScaledGD). The proof of the rank-d case follows the sketch of the rank-1.
>
> Specifically, the error $ || U_k V_k^{\top} - M||_F $ is upper bounded by four terms in ScaledGD as in Eq. (12) and three term in AltScaledGD as in Eq. (25), all the remaining lemmas and inequalities are to guarantee that each of the upper-bound terms decrease linearly. To this end, the inequality $ <U_k V_k^{\top}, M > \geq || U_k  V_k^{\top}||_F^2$ is crucial to our analysis since once it is satisfied, according to Lemma 4, Lemma 5 and Lemma 6 the upper bound terms in Eq. (12) and Eq. (25) decrease linearly, which reveals the linear convergence of ScaledGD and AltScaledGD.
>
> Once again, we thank the reviewer for the valuable and constructive suggestion. We will revise our manuscript of section 4 and the Appendix in the revision such that the overall picture of the proof can be clear and easy to follow.
>
> **Q3: Some typos need to be rectified and the writings need to be improved.**
> **A3**: We will carefully proofread the entire manuscript to ensure that all the writing mistakes will be rectified.
>
> **Q4: Where is the dependence of $\sigma$ in Theorem 1 statement?**
> **A4**: Thank you for your careful reading and pointing this out. $\sigma$ determines the scale of the initialization and correspondingly the initial error $||U_0 V_0^{\top} -  M||_F$, which further determines the time the initial phase lasts. Specifically, the variable $\sigma$ is contained in $T_1$ as $T_1 = O(\ln \frac{\sigma d}{\delta})$, since $\sigma$ is only a constant ($O(1)$) compared to the problem dimension and other parameters, we thus omitted it in our previous submission. We will revise the corresponding part in the revision.

---

> > ### Comment · Reviewer_QnG3 · 2023-08-14
> > **Acknowledgment.**
> >
> > Authors answer my questions sufficiently and promised to address the changes in later revisions. Another point is that they promised that that the constants will be made more explicit in the paper. With those in mind, I do not mind raising my score by one.

---

> > > ### Author Response · Authors · 2023-08-15
> > >
> > > We are genuinely grateful for your time and effort to reviewer our work and response to our rebuttal. It is with your the help of your thoughtful review that we have been able to enhance the quality of our work, ensuring that it meets or exceeds the expected standards. Your attention to detail has helped us refine our writing and bring greater clarity to our message.

---

### Author Rebuttal · Authors · 2023-08-07

Dear AC and reviewers,

Thank you so much for your valuable comments and we truly appreciate the time and effort you've taken to review our work. We are also glad that the reviewers found our work valuable and give us positive feedbacks. Your feedback is very important to us, and according to the reviewers comments, we have revised the manuscript carefully and made detailed response to all your concerns (**please refer to our rebuttal for each reviewer for more detailed responses**).

As this paper focus on the theoretical side to prove the global convergence of ScaledGD and its variant AltScaledGD on the basic model of the low rank matrix recovery problem, reviewers may concern whether ScaledGD/AltScaledGD can be used for more advanced non-convex optimization problem such as nonlinear neural networks. To use ScaledGD/AltScaledGD for the optimization of deep learning is a very interesting research direction, we are still working on this problem. We provide one page PDF file to show some experimental results on this problem. Specifically, we show in the attached PDF file Figure 1 and Figure 2 the experimental results of ScaledGD/AltScaledGD on deep linear network and non-linear network, which all certificates that ScaledGD/AltScaledGD converge much faster than vanilla GD. Though ScaledGD/AltScaledGD work well for deep network, their convergence analysis require new theoretical tools, which is quite challenging compared to the convergence of LRMF, we will present the results in our next work.

Meanwhile, we also tested the ScaledGD/AltScaledGD on real data sets in Figure 3 and Figure 4 of the PDF file. In real data sets, the target matrix is not exactly low-rank but approximately low-rank with very large condition number $\kappa$. The experimental results show that ScaledGD/AltScaledGD still work well for real data sets compared to GD.

Though the proof of our results is a little bit lengthy, and requires some important lemmas and inequalities, we have provided the overall picture of the proof by the proof sketch for rank-1 case in section 4. The proof for rank-1 case is easy to follow and understand. The proof of rank-d follows directly the proof sketch of rank-1 but with moderate changes of lemmas and inequalities from rank-1 to rank-d, we thus leave the proof in Appendix. In this paper, we first present the main results in section 3 and provide an overall picture of the proof in section 4, detailed proof are provided in the Appendix, therefore the overall structure of the manuscript is clear. In the revision, we will further improve the organization of this paper to make it easy to read.

Once again, thank you for taking the time to review our work and providing valuable insights, and we will take all the suggestions given by the reviewers into consideration for future improvements. We would also like to have discussions with all the reviewers if you have any further concerns.

---

### Decision · Program_Chairs · 2023-09-21

**Decision:**

Accept (poster)

**Comment:**

The reviewers agree that the paper makes a solid, worthy contribution. One dissenting reviewer expressed some skepticism with the rank-1 proof for ScaledGD. Indeed, one can argue that in the rank-1 case, ill-conditioning is impossible, and this completely obviates the whole point of preconditioning. I would also very much like to see the minor revision done for the rank-d for ScaledGD. Please incorporate reviewer feedback into the final submission.